# Array-Based Epigenetic Aging Indices May Be Racially Biased

**DOI:** 10.3390/genes11060685

**Published:** 2020-06-22

**Authors:** Robert Philibert, Steven R.H. Beach, Man-Kit Lei, Frederick X. Gibbons, Meg Gerrard, Ronald L. Simons, Meeshanthini V. Dogan

**Affiliations:** 1Department of Psychiatry, University of Iowa, Iowa City, IA 52242, USA; meeshanthini-vijayendran@uiowa.edu; 2Behavioral Diagnostics LLC, Coralville, IA 52241, USA; 3Center for Family Research, University of Georgia, Athens, GA 30602, USA; srhbeach@uga.edu (S.R.H.B.); karlo@uga.edu (M.-K.L.); ron.lee.simons@gmail.com (R.L.S.); 4Department of Psychological Sciences, University of Connecticut, Storrs, CT 06268, USA; rick.gibbons@uconn.edu (F.X.G.); meg.gerrard@uconn.edu (M.G.)

**Keywords:** epigenetics, DNA methylation, epigenetic aging, healthcare disparities

## Abstract

Epigenetic aging (EA) indices are frequently used as predictors of mortality and other important health outcomes. However, each of the commonly used array-based indices has significant heritable components which could tag ethnicity and potentially confound comparisons across racial and ethnic groups. To determine if this was possible, we examined the relationship of DNA methylation in cord blood from 203 newborns (112 African American (AA) and 91 White) at the 513 probes from the Levine PhenoAge Epigenetic Aging index to ethnicity. Then, we examined all sites significantly associated with race in the newborn sample to determine if they were also associated with an index of ethnic genetic heritage in a cohort of 505 AA adults. After Bonferroni correction, methylation at 50 CpG sites was significantly associated with ethnicity in the newborn cohort. The five most significant sites predicted ancestry with a receiver operator characteristic area under the curve of 0.97. Examination of the top 50 sites in the AA adult cohort showed that methylation status at 11 of those sites was also associated with percentage European ancestry. We conclude that the Levine PhenoAge Index is influenced by cryptic ethnic-specific genetic influences. This influence may extend to similarly constructed EA indices and bias cross-race comparisons.

## 1. Introduction

Over the past 10 years, there have been significant advances in our ability to use epigenetic status to predict healthcare outcomes. Progress in this area has been facilitated by the development of methylation profiling platforms, in particular the Illumina Human-Methylation450 BeadChip (450K) and Infinium MethylationEpic Beadchip (Epic) arrays that are capable of assessing methylation status at hundreds of thousands of CpG residues simultaneously [1,2].

In the world of research, these arrays have been used for two distinct purposes. First, they have been used for discovery. Using these arrays, literally thousands of studies describing the relationship of methylation to illnesses, such as diabetes and heart disease, or to environmental exposures/consumption of substances such as cigarettes, alcohol, and pesticides have been conducted [3,4,5,6,7,8,9]. Importantly, the findings from many of these studies have been subsequently replicated in cohorts of other ancestries thus ensuring the generalizability of the findings across ethnic groups. Additionally, array-based methylation assessments have often been verified by more exact approaches for assessing methylation such as pyrosequencing [10].

These arrays have also been used as tools for imputing global health status. Perhaps the most popular of these metrics is that of “epigenetic aging” (EA). Although the concept owes its origins to a variety of individuals including Fraga and Esteller, the implementation of the information from these arrays as EA indices was described separately and independently by both Hannum and associates and Horvath and associates in 2013 [11,12,13]. At the core of their approaches, each team first used regression to identify a set of several hundred CpG loci whose methylation status changes, either increasing or decreasing, in association with chronological age in a first set of subjects. Then, they used the rate of change at each of these loci in array data from a second group of subjects to predict the apparent chronological age of the subjects with the difference between the apparent and actual chronological age being referred to as “accelerated epigenetic age”. 

Given the social and economic challenges of the burdens associated with aging, the use of these global indices has been remarkably popular. Hundreds, if not thousands of papers have used these indices, or some derivative of them to predict key healthcare outcomes associated with aging such as mortality and cardiovascular disease. More recently epigenetic indices have been developed to predict disease phenotypes. In particular, the “PhenoAge” DNA methylation index developed by Levine and colleagues was designed to overcome limitations of age focused prior measures [14]. Because EA indices keyed to chronological age were not found to be consistently related to cardiovascular disease or early onset of chronic illness [15], DNAm PhenoAge was developed using both chronological age and clinical measures so that it would better predict individual differences in lifespan and health span [16] The index reflects several known aging pathways and provides a useful objective marker of elevated risk for early onset morbidity and chronic illness. A potential limitation of the DNAm PhenoAge index, and perhaps all similarly formed indices, is the extent it was trained on data containing ancestry specific methylation information. If the ethnic groups whose data are represented in the training sets are mismatched for socioeconomic factors that also affect health outcomes, the resulting algorithm may inadvertently be biased, resulting in an index that is not well suited for comparisons across race, or for application to broad, heterogeneous samples. 

The concern regarding possible contamination by cryptic ethnic variation in EA indices is not hypothetical. It is well-established that methylation arrays contain significant implicit and explicit genotype information [17,18,19]. To understand how implicit genotype information becomes detectable on methylation arrays, it is necessary to recall that the key difference between the arrays that measure genetic as compared to those which measure epigenetic variation is that while both arrays quantitatively capture allele specific hybridization signals, the former uses regular genomic DNA while the latter uses bisulfite converted DNA. Because only a portion of the sequence variation is destroyed by bisulfite conversion, it is possible to assess explicit and implicit genetic information from bisulfite converted DNA. The 450 K array provides explicit genotype information at 65 genetic polymorphisms whose genotypic variation (e.g., an A to G polymorphism) is unaffected by the bisulfite conversion [20]. These 65 genotypes on the array can be used to help sort out any laboratory mix-ups with respect to subject identification. The implicit genotyping information is considerably larger and takes advantage of the fact that many of the 50 bp pair long probes hybridize to segments of the genome containing not only a CpG residue, but genetic polymorphisms as well. Because the presence of the polymorphism can directly alter the annealing temperature of a segment of DNA for a given C or T specific probe or indirectly by changing the amount of methylation at a given site, the presence of the genetic polymorphism can be inferred. The presence of many of these potential genetic influences is noted in the 450K and Epic annotation files [20]. In 2014, we showed how “genetic” information contained in the hybridization signal from the 450K analyses of 111 African Americans (AAs) subjects could be used to infer over 10,000 genotypes [17]. However, each ancestry will have its own unique set of signals with the number of potential detectable gene-methylation interaction effects likely to be in the millions [21,22].

Relevant to our current investigation, many cryptic genotypic influences are ethnically specific. In fact, Rahmani and colleagues have developed a program called EPISTRUCTURE for inferring ethnicity from the cryptic genotyping information [18]. This cryptic variation may affect important health related loci. For example, with respect to smoking, we have shown that almost 90% of the top ranked CpG sites for predicting are affected by ethnic specific genetic variation [23]. In the Framingham Heart Study (FHS) Offspring Cohort, which consists of individuals of European ancestry, 195 of the 513 CpG sites in the DNAm PhenoAge EA Index are significantly associated with methylation at the well-established indicator of smoking, cg05575921 [24]. However, in AA samples, the Levine PhenoAge Index is only modestly associated with cotinine seropositivity and cg05575921 inferred smoking intensity [25]. 

Might discrepancies in the pattern of correlation between DNAm PhenoAge EA and health behaviours for those of differing ethnicity be the result of differences introduced by unacknowledged cryptic genetic influences? Despite being based on methylation platforms that are designed to capture acquired changes, both the DNAm PhenoAge index and newly described GrimAge have strong heritable components. The DNAm PhenoAge index has a h^2^ of 0.51 while the GrimAge index has a h^2^ of 0.37 with the individual subscales have heritabilities between 0.34 and 0.51. [14,26] Conceivably, part of the heritability could come from germline transmission of methylation signals. However, although epigenetic inheritance has been posited in humans, none has conclusively demonstrated with most experts agreeing that if it occurs, the amount of germline transmission is fairly low [27,28]. Accordingly, it is important to carefully examine the source of any observed heritability of EA indices and determine whether ethnic specific components in that heritability affects are correlates.

In this communication, we hypothesize that some of the inconsistency across samples of differing races/ethnicities when examining associations of the DNAm PhenoAge EA Index with health behaviors; and, some of its heritability, may be the consequence of methylation being affected by ethnic specific genetic variation. To test this hypothesis, we examine the association of methylation values at individual probes from the DNAm PhenoAge EA index with ethnicity in methylation data from two separate cohorts. The first set of data is from a set of 203 newly born infants (112 AA and 91 White). We then tested and extended the most significant finding to a cohort of AAs for whom we can infer varying degrees of European ancestry. 

## 2. Materials and Methods 

The genetic and epigenetic data used in this study were obtained from two sources. The list of the probes used in the Levine EA Index was taken from their 2018 work [14]. 

The first set of data was from a set of methylation assessments of newborn cord blood DNA conducted by Mozhui and associates as part of their 2015 study of maternal nutrition [29]. In brief, after obtaining approval from the University of Tennessee Institutional Review Board (IRB 200802719), Mozhui and colleagues obtained cord blood from 212 newborns from the University of Tennessee Health Center including samples from 112 AA and 91 White subjects. Methylation array profiling of the DNA samples prepared from these samples was conducted using the Illumina Humanmethylation27 BeadChip and processed using the Illumina Genome Studio (version 2009.1) [29]. The data were then corrected for batch effects using the COMBAT R package. The resulting M-values for 27,577 probes (including all 513 probes from the DNAm PhenoAge EA index) for all 212 samples were then generously posted to the NCBI NIH Gene Expression Omnibus (accession ID GSE64940).

The source and processing of the genome wide epigenetic data from the Family and Community and Health Studies (FACHS) have also been previously described [30,31]. In brief, the FACHS study was a longitudinal study of the effects of socioeconomic factors on health-related outcomes of AA parent-child dyads from Iowa and Georgia. During Wave 5 of this longitudinal study (2008–2010), the adult subjects from these dyads were interviewed and phlebotomized. The DNA from these samples were processed via our standard procedures, then interrogated for genome wide methylation using the Infinium MethylationEpic Beadchip by the University of Minnesota Genome Center. Standard sample and probe level quality control were conducted as previously described [23,25]. After quantile normalization, the resulting methylation values were exported as beta values for use in this study.

The genetic information from FACHS cohort used in the ancestry index described below was obtained using Infinium Multi-Ethnic Global-8 Beadchip by the University of Minnesota Genome Center. In short, after processing with Genome Studio, the data were subjected to quality control measures at both the sample and SNP probe levels using PLINK [32]. Subject data from whose self-reported gender and biological sex were discordant or whose heterozygosity rate was greater or smaller than the mean ± 2SD and with a proportion of missing SNPs > 0.03 were excluded. 

The European Ancestry index (EAI) was constructed using a list of ancestry informative polymorphisms identified by Seldin and associates [33]. In brief, in 2009 this group identified a set of 128 genetic markers whose genotype status could be used to infer the ancestral origins of subjects of anyone in the world. From their list of 128 single nucleotide polymorphisms (SNPs), we selected 13 (rs9809104, rs385194, rs6556352, rs2504853, rs1871428, rs7803075, rs2416791, rs772262, rs9522149, rs4984913, rs2125345, rs7238445, rs4891825) whose major and minor allele frequencies were markedly discordant between those with African vs European ancestry (e.g., 10% A in one population and 90% A in the other population) and whose data were available from the Multi-Ethnic Global-8 Beadchip. We then converted their genotypes at each of those SNPs to a 1, 2 or 3 scale with 1 being the genotype most common in those with African ancestry and 3 being the genotype most common with European ancestry (e.g., AA = 1, AG = 2 and GG = 3 for rs385194). The scores at each these loci were averaged to provide an EAI score for each subject.

All data were analyzed using the JMP suite of programs (Cary, SC) using the statistical tests (T-tests, logistic regression, and receiver–operator characteristic (ROC) area under the curve (AUC)) analyses described in the text [34,35]. 

## 3. Results

As a first step in understanding the potential for ethnically contextual genetic effects affecting methylation status at the 513 loci used in the DNAm PhenoAge EA index, we analyzed the relationship of methylation score to ancestry using the cord blood DNA methylation data from 112 African American (AA, 59 male and 53 female) and 91 White (41 male and 50 female) subjects profiled by Mozhui and associates [29].

In total, methylation status at 223 of the 513 probes in the Levine EA index were nominally associated with ethnicity with methylation status at 50 of these probes being significantly associated (*p* < 0.05; *t*-test) after Bonferroni correction. Table 1 lists the 30 probes whose methylation status is most significantly associated with ancestry along with information regarding the presence of nearby polymorphisms from the 2016 edition of the Illumina Human-Methylation450 BeadChip annotation file. A listing of the complete association analyses is given in Appendix A. 

Although the genetic variation potentially affecting DNA methylation status at any given residue can be anywhere in the genome, genetic variation immediately adjacent to the CpG site is thought to have particularly strong effects [36]. Review of the annotation information from Illumina (see Table 1) shows that 7 of the 30 most significant probes have known polymorphisms within 50 bp of the CpG site with yet an eighth probe (cg12864235) having a SNP (rs73925316) within 10 bp of the CpG site specifically interrogated by the probe. Examination of that locus in the NCBI dbSNP database [37] shows a marked discrepancy between the allele frequencies between Africans and Europeans with the frequency of the G allele being 0.0484 (*n* = 2072) in Europeans and 0.3 in AAs (*n* = 76). Review of the dbSNP data at the other less closely located SNPs from Table 1 show similar ethnic specific differences in several of the 7 SNPs.

To understand the power of methylation values at these CpG sites to predict ethnicity, we conducted a series of nominal logistic regression analyses. A simple model just using the information from the most highly associated CpG probe (cg08654655) was highly significant (*p* < 0.0001, R^2^ = 0.225, ROC AUC 0.80, *n* = 203). Stepwise addition of methylation information of the next 4 most highly ranked probes steadily increased the predictive power for ethnicity status to (*p* < 0.0001, R^2^ = 0.66, ROC AUC 0.97, *n* = 203). In contrast, although ethnicity had a strong effect on methylation, there was no effect of gender in any of the models.

As a next step of our analyses, we tested whether methylation at the most significantly associated sites was associated with the percentage of European ancestry in our adult AA cohort. As a first step, we calculated an index of the relative amount of European ancestry using the method outlined by Seldin and associates. As Figure 1 shows, the relative amount of European ancestry varied widely. The average value of the European Ancestry Index (EAI) was 1.36 with 15 self-reported AA subjects averaging at least one or more of the alleles normally associated with Europeans across each of the 13 polymorphic sites surveyed.

We then analyzed the relationship of the EAI to methylation status at each of the 50 CpG sites that were significantly associated with ethnicity in the newborn cohort. Overall, 11 of the 50 CpG sites associated with AA ancestry among infants were also associated with the degree of AA ancestry in the adult FACHS subjects (see Table 2). Interestingly, five of the eleven CpG probes were among those associated with cg05575921 status in the prior study of the DNAm PhenoAge Index using the Framingham Heart Study population. 

## 4. Discussion

Using methylation and ethnicity data from two cohorts, we found that methylation status at some of the sites used in the DNAm PhenoAge index is associated with ethnicity. Caveats include the relatively limited power of the confirmation cohort, possible confounding biases in the newborn population, and the possibility that some observed differences may be attributable to differential prenatal exposures. Finally, because the UCLA website (www.http://dnamage.genetics.ucla.edu/) that calculates the DNAm PhenoAge EA does not accept 27K data for imputation of the DNAm PhenoAge Index, we were unable to directly compare the averages and distribution of the DNAm PhenoAge EA predictions for these subjects.

Although perhaps ungainly at first consideration, the study design purposefully addressed the possibility of racial/ethnic confounding by using cohorts with different types of contrasts. In the first cohort, the contrast was between subjects. All of the blood samples came from the same source, cord blood, and at the same age, birth. Hence, there were no confounds with respect to age or DNA source making this an intuitive, easy to understand approach to identifying potential ethnic specific methylation variation that minimizes other potentially confounding influences. At the same time, this case and control approach is subject to other biases, such as the potential for White mothers to have greater access to pre-natal care or different prenatal experiences. Conceivably, these differences could be reflected in methylation differences among infants. Arguing for a genetic source for these differences, however, we note that local genetic variation is documented in 8 of the 30 most significantly affected probes. So, there is good reason to believe that genetic factors may already be in play at these loci at birth, and that some of the findings represent differences due to cryptic ethnic genetic variation rather than differential prenatal experience. Such differences have the potential to yield scales that reflect different influences across racial/ethnic groups. For example, in the FHS (White) sample, the DNAm PhenoAge EA index largely loaded on cigarette and alcohol consumption and this was not the case in the FACHS (AA) sample [24,25]. Since the rate of smoking and alcohol consumption are roughly equal in Whites and AAs [38,39], it is unlikely that differences in habits affected the methylation outcomes, suggesting that the DNAm PhenoAge index is operating somewhat differently across ethnic groups. 

In the second set of analyses, the contrast was within individual- using individuals who varied by age. Still, since all of these individuals are from the same cohort, the likelihood that some systematic bias in nutrition, substance use, community, or socio-economic adversity is differentially affecting individuals as a function of their EAI score is limited, although not impossible. More likely is that the EAI predicts the proportion of ethnic specific genetic variation that can raise or lower the methylation particular loci. Despite the limited number of subjects with high EAI scores, methylation status at 11 of the loci was significantly associated with EAI which suggests that a larger more informative cohort could have confirmed the associations at additional loci that did not achieve statistical significance in the current sample. Unexamined was the extent to which EAI was associated with sources of stress, including economic hardship and discrimination, that might also contribute to the association.

A reasonable question is how accurate is the EAI approach that we used in this study. Certainly, the use of additional ethnically informative SNP information would provide more precise estimates. Unfortunately, of the 24 SNPs most highly predictive of White versus AA ancestry in the 2009 manuscript by Seldin and colleagues, only 13 were included in the Global-8 Multi-Ethnic chip. Still, we note that all things being equal, the average value of 1.36 suggests that, on average, 18% of the ancestry of FACHS participants is of European origin. This figure is well in line with prior estimates of 24% by Bryc and associates in their analyses of 5269 self-reported AA subjects [40].

The fact that 5 of the CpG sites whose confounding by ethnic specific genetic variation was confirmed in the second set of analyses were also significantly associated with epigenetic smoking in prior studies of the FHS is not unexpected. Recently, we reported a set of analyses that showed that cg05575921 and a GrimAge sub-index (packyears), but not the DNAm PhenoAge EA index strongly predicted smoking status in the FACHS cohort [25]. The finding that the methylation signal at these loci is affected by ethnic specific variation supports the assertion that confounding may have diminished the association between the DNAm PhenoAge EA index and health behaviour in the FACHS cohort relative to white samples. We also note that none of these sites were significantly associated with smoking in any of our two prior genome wide studies of smoking in AAs [23,41].

The findings in this manuscript further emphasize the need to understand the genomic basis for the genetic x methylomic (i.e., GxMeth effects) interactions. The existence of these GxMeth effects has been known for many years. In 2010, Mill and colleagues noted widespread allele specific methylation skewing [42]. In 2013, Illumina posted a product note to their website that indicated that over half of the (273,660 of the 485,577 total) probes in the 450 K array had one of more significant GxMeth interaction effects [43]. Since then, a number of studies have shown that the genetic variation detected by these platforms is widespread, has significant impact, and can be used to predict ethnicity [18,23,43,44]. Still, it is important to note that these arrays only assess a fraction of the 28 million CpG sites in the human genome [45]. Therefore, whole genome bisulfite sequencing approaches will be needed to establish a comprehensive understanding of the large and fine scale methylomic regulation. If successful, these studies may lead to the development of precision medicine tools for the treatment of certain developmental disorders which result from defective genomic imprinting or cancer [46].

What is not clear from this study is what proportion of the signal in the DNAm PhenoAge EA index, loads on ethnicity. In part, that is because the process involved in its derivation is complex and relies on data from a number of cohorts each of which possess unique ascertainment and demographic characteristics. Still, the results from the current study combined with the fact that 51% of the DNAm PhenoAge EA index is reported to be heritable, provide a conceptual challenge to the hope that it would provide a relatively pure index of the impact of acquired effects of the environment on the epigenome. 

Are other recently developed EA indices that use methylation arrays also subject to ethnic specific genetic effects? The Brenner Frailty index bases its prediction on the information from a relative handful (34) of CpG sites [47]. If the amount of confounding is proportional to the number of probes, then the effects on this index are likely less severe. In addition, many of the CpG sites in their index, such as cg04987734 and cg05575921, load specifically on smoking and drinking consumption and have no known ethnic bias in their set points [48,49]. In contrast, the GrimAge index, which also has considerable heritability, uses information from over 1000 CpG probes [26]. However, the identity of those probes has not been publicly disclosed, making examination of contamination by cryptic ethnic variation difficult.

The policy implications of this study are potentially significant. In essence, we show that a number of loci the DNAm PhenoAge index can be used to predict ethnicity and thereby that some of its prediction of health outcomes in general population samples may be secondary to health disparities between Blacks and Whites or other ethnic groups. Concerns about potentially different patterns of correlates are also relevant to health advice based on these measures. For example, companies using EA indices commercially to provide estimates of biological age and indicate the need for products such as vitamin supplements [50], may need to reconsider the quality of advice being provided to non-Whites. Because supplements are relatively non-toxic, this may be of minor concern at present. However, use of these or similar tools in more extensive medical decision making could be more consequential. Likewise, if these tools are used to guide changes in Federal or State policy, or allowed to influence insurance costs, the financial consequences for groups showing higher scores due to contamination by cryptic ethnic variation could be substantial. 

## 5. Conclusions

In summary, we report that the DNAm PhenoAge EA index contained some ancestry specific information. Although measures of EA are useful in a variety of research contexts, particularly when focused on homogeneous samples, we suggest the need for caution in the use of this and similar tools in situations that explicitly or implicitly involve comparisons across racial or ethnic groups. It is possible that effects of contamination by cryptic ethnic variation are limited to main effects that could be corrected statistically. Conversely, there is reason to worry that effects of contamination by cryptic ethnic variation may also extend to patterns of association. In that case, observations and recommendations regarding predictors and consequences of EA measures may need to be carefully replicated with multiple ethnic groups to directly test the extent of generalizability. Alternatively, efforts to ensure that EA measures are free of contamination by cryptic ethnic variation, equally applicable to multiple ethnic groups, and responsive to similar predictors across ethnic groups may require some revision of the EA measures that are currently in widespread use. 

## Figures and Tables

**Figure 1 genes-11-00685-f001:**
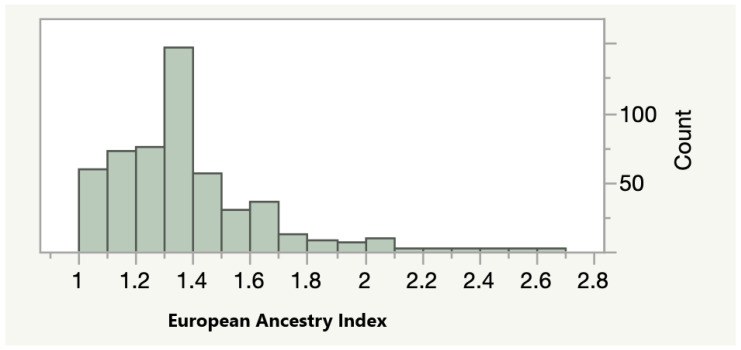
The Distribution of the European Ancestry Index (EAI) in the FACHS Adult Subjects (*n* = 505). A score of 1 indicates the presence only alleles enriched in African subjects while a score of 3 indicates the presence of alleles enriched in European subjects.

**Table 1 genes-11-00685-t001:** The thirty probes most significantly associated with ancestry.

Illumina Probe ID	*t*-Tests	BF Corrected	CHR	SNPs within 50 bp *	SNPs within 10 bp **
cg08654655	2.28 × 10^−15^	1.17 × 10^−12^	1		
cg18771300	3.13 × 10^−15^	1.61 × 10−^12^	14		
cg15344028	9.86 × 10^−15^	5.06 × 10^−12^	2		
cg02016419	2.05 × 10^−12^	1.05 × 10^−9^	17		
cg12402251	4.56 × 10^−12^	2.35 × 10^−9^	8		
cg04718414	5.78 × 10^−12^	2.96 × 10^−9^	13	rs17337675	
cg08251399	1.81 × 10^−11^	9.29 × 10^−9^	2		
cg12864235	2.6 × 10^−11^	1.34 × 10^−8^	5		
cg09799873	2.26 × 10^−10^	1.16 × 10^−7^	19		rs73925316
cg00862290	4.37 × 10^−10^	2.24 × 10^−7^	3		
cg06638451	5.77 × 10^−10^	2.96 × 10^−7^	3	rs17059410	
cg19566405	7.70 × 10^−10^	3.95 × 10^−7^	17		
cg13509147	7.77 × 10^−10^	3.99 × 10^−7^	19		
cg20066677	1.10 × 10^−9^	5.62 × 10^−7^	12		
cg16713727	3.18 × 10^−8^	1.63 × 10^−5^	1		
cg11618577	3.49 × 10^−8^	1.79 × 10^−5^	2		
cg12813792	4.94 × 10^−8^	2.53 × 10^−5^	20		
cg04836038	7.23 × 10^−8^	3.71 × 10^−5^	13		
cg27187881	8.21 × 10^−8^	4.21 × 10^−5^	22		
cg10795646	9.29 × 10^−8^	4.77 × 10^−5^	1		
cg15201877	2.32 × 10^−7^	0.0001191	1		
cg17133388	4.10 × 10^−7^	0.0002104	3		
cg13119609	4.64 × 10^−7^	0.000238	19		
cg22736354	4.96 × 10^−7^	0.0002545	6	rs28940575	
cg23159337	8.61 × 10^−7^	0.0004416	3	rs34959916	
cg24125648	1.21 × 10^−6^	0.000618	15	rs75056397	
cg15963417	1.38 × 10^−6^	0.0007092	12	rs62652660	
cg09304040	1.58 × 10^−6^	0.0008097	12		
cg09404633	1.82 × 10^−6^	0.0009359	1		
cg10570177	2.56 × 10^−6^	0.0013111	9	rs36223203	

BF: Bonferroni, * and ** refer to the presence of polymorphisms with 50 and 10 base pairs of the CpG targeted by the probe.

**Table 2 genes-11-00685-t002:** The association of methylation at the 11 most significantly associated probes with EAI Score.

ID	*p*-Value	Associated with Smoking in FHS ^a^
cg06638451	0.0002941	No
cg04718414	0.0004285	Yes
cg00168942	0.0008096	Yes
cg00862290	0.0035711	No
cg10795646	0.0104504	No
cg19514469	0.0115081	Yes
cg08251399	0.0149561	Yes
cg09404633	0.0241233	No
cg08067365	0.0354048	No
cg07038400	0.040688	No
cg03991512	0.0426108	Yes

^a^ As shown in Mills et al. (2019) [24]. FHS: Framingham Heart Study.

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
