# Peer review of "Array-Based Epigenetic Aging Indices May Be Racially Biased"

_genes, 2020, doi:10.3390/genes11060685_

Round 1
Reviewer 1 Report
DNA methylation-based indices of health behaviors, aging, drug use have been increasingly accepted and proved to be accurate. This article is aiming to examine an aspect of this diagnostic entity in relation to the potential bias based on race-related genetic variations. The authors quote an example for this hypothesis, where the DNAm PhenoAge EA Index is significantly associated with methylation at the well-established indicator of smoking in a study of individuals of European ancestry, but only modestly in an AA population.
- One of the selected studies, the cord blood one, shows association of the 513 probes of the Levine EA index with ethnicity, but only the top 30 probes and some of the nearby polymorphisms are shown. It would be more informative to show the nearby polymorphisms for all the significant (50) and nominally significant (223) probes with allele frequencies by ethnicity.
- Please spell out in the abstract what is the Levine EA index of ethnicity
- Introducing the GrimAge index early in the introduction is important if the discrepancy between the heritability difference between DNAmPhenoAge and GrimAge is relevant.
- Accidental inclusion of the sentence “Interventionary studies involving animals or humans, and other 167 studies require ethical approval must list the authority that provided approval and the corresponding 168 ethical approval code” at the end of methods.
- The authors found no gender effects in methylation which seems to be in contrast to those reported by Horvath et al., 2016.
Author Response
Dear Ms. Rubio:
Please find the accompanying revised manuscript entitled ”Array Based Epigenetic Aging Indices May Be Racially Biased.” The article received two reviews. The comments from the first reviewer are listed below.
Reviewer 1.
Comment: “It would be more informative to show the nearby polymorphisms for all the significant (50) and nominally significant (223) probes with allele frequencies by ethnicity”
Response: We agree and have edited Supplemental Table 1 to show all known local SNPs as denoted by Illumina. The Reviewer will understand that listing all SNPs that affect methylation, both cis (defined as a bin of 500 kb on either side of the site) and trans, by ethnicity, is an impossibility (see Dogan et al., 2016). Even a CpG site with no known ethnic effects on set point, such as cg05575921, has a large number of significant GxMeth interactions. It is just that in general, in all ethnicities tested, these effects are still minor compared to the degree of environmentally induced change and tend to cancel one another out. This is not so for the sites identified in this study. Please see the extensive response that we have given for this subject with respect to Reviewer 2’s comment.
Comment: “Please spell out in the abstract what is the Levine EA index of ethnicity”
Response: We have made the change suggested by the Reviewer. In particular, we specifically used the PhenoAge term used by Dr. Levine and her colleagues.
Comment: “Introducing the GrimAge index early in the introduction is important if the discrepancy between the heritability difference between DNAmPhenoAge and GrimAge is relevant”
Response: The discrepancy between the heritability probably isn’t that significant to understanding the problem being addressed. Indeed, since the probes in the GrimAge index are not publicly known, it is difficult to say anything other than what Lu et al. (from the Horvath group) has written.
Comment: “Interventionary studies involving animals or humans, and other 167 studies require ethical approval must list the authority that provided approval and the corresponding 168 ethical approval code”
Response: We have no idea where that text originated. I wonder if it is from the template. In any case, we have deleted it.
Comment: “The authors found no gender effects in methylation which seems to be in contrast to those reported by Horvath et al., 2016”
Response: The “no effect of gender” comment only refers to the model using those five probes. I am sure if we looked at all of the probes, we would find some effects. As a side note, we can tell from our other analyses that some of those in the GrimAge index, map to sex chromosomes.
Comment: They tentatively suggest caution in using methylation-based epigenetic scores in the same way in African American vs European American samples.
Response: By nature, guided by the principle “do no harm,” we are inherently conservative in our approach for the use of new tools. Hence, we routinely offer caution. In this case, we think that caution is well merited.
Reviewer 2 Report
Philbert and colleagues use a modest number of genomic SNP genotypes to infer overall genomic ancestry, and then find that the results of bisulfite DNA/array based genomic methylation assays differ at sites that have nearby SNPs as well as those that do not. They tentatively suggest caution in using methylation-based epigenetic scores in the same way in African American vs European American samples.
The authors fail to acknowledge the opportunities presented here: to infer local genomic ancestry (based on SNP genotypes) for each of the methylation sites of especial interest. It would be a straightforward task to SNP genotype these samples (if not done already), construct local ancestry estimates for each genomic DNA segment that contains a methylation site of interest, and use this local ancestry data to control for extent of methylation at each site. The genomic admixture in African-Americans provides a golden opportunity for analyses based on local ancestry.
As the paper stands, it provides a much more modest advance that is likely to be much less of an advance for the field. If published based on global ancestry estimates, it should be substantially shortened with much attention to the looseness of language in introduction and discussion. However, I would strongly urge attention to local ancestry in analyses that would dramatically improve the field of epigenetics.
Author Response
Dear Ms. Rubio:
Please find the accompanying revised manuscript entitled ”Array Based Epigenetic Aging Indices May Be Racially Biased.” The second reviewer really only had one comment that was reiterated several times. We respond to in length and by adjusting the discussion. Our point by point responses to his/her overlapping comments are listed below.
Reviewer 2.
Comment: The authors fail to acknowledge the opportunities presented here: to infer local genomic ancestry (based on SNP genotypes) for each of the methylation sites of especial interest.
Response: The Reviewer makes an interesting point and one in which we are strongly invested. Conceivably, by using whole genome bisulfite sequencing, one could identify local SNPs associated with ethnic specific differences. However, given the number of haplotypes and the short read inherent in bisulfite sequencing, it would be a heroic task to attempt.
Still, identifying ancestry specific SNP effects is not as straightforward as it sounds. For example, we are preparing to submit a paper that uses GxMeth effects to improve an existing Illumina methylation biomarker for a common illness using the information that very FACHS cohort (we have the clinical and biomaterial from both the parents and the children in our biobank-this includes genome wide methylation and genotype data). The local haplotype to which this CpG probe featured in this analysis maps has several dozen SNPs, each of which produce, to a certain extent co-linear, effects on prediction. However, these local effects on prediction are dwarfed by the 140 genome wide significant trans GxMeth interaction effects. Obviously, we are using machine learning approaches to discern the most effective combination of SNPs to pair with the probe (right now, the local SNPs account for 2% of the variance; trans SNPs account for 25% of the variance). Further, I think the Reviewer will be impressed by our grasp of this subject in the paper describing our new integrated genetic-epigenetic algorithm for predicting heart disease that Dr. Dogan, in collaboration with Intermountain Healthcare, is submitting for publication.
But the reality is that as someone who has used (and published) EWAS, GWAS and integrated genetic-epigenetic approaches to study both African American and White populations is it is clear to me that virtually the entire variable epigenome is subject to these ethnic specific effects. As early as 2013, (almost two years after introducing the chip), Illumina realized this point and provided a list of 273,660 probes (half of the array) with one or more significant SNPxMeth interaction effects. These GxMeth effects can be exposure specific. In Dogan et al. 2017 we showed that after Bonferroni correction, just with respect to smoking there were 827 and 448,342 significant cis‐ and trans‐interactions (representing nearly 79,000 unique SNPs) in the FHS. The vast majority of the SNPs in this list of 79,000 SNPs show ethnic specific effects. But there already is a vast reservoir of knowledge in this area. Jonathan Mills is among those way out ahead of the field in this regard. In 2016, he started an ancestry specific methylation data base (https://epigenetics.essex.ac.uk/ASMBrainBlood/) and showed effects at 4% of the 220,000 loci whose methylation he examined in brain tissue. Since then, given the number of data sets that are coming on line, the number of loci and haplotypes that have been contributed to his database has probably grown greatly.
I think that the Reviewer has made an interesting point and one that is essential if we are to bioengineer the genome to mitigate disease and improve quality of life. But if we were to do this, it should be done systematically and using a study design powered to do this (500 AA and 500 White subjects with paired genome wide multiple tissue methylomic and genetic data.
Still, the Reviewer notes a limitation of the paper, and we have added text to the discussion to note this, and to reference others working on this important matter.
Comment: As the paper stands, it provides a much more modest advance that is likely to be much less of an advance for the field……., I would strongly urge attention to local ancestry in analyses that would dramatically improve the field of epigenetics.
Response: Please see our above response to this important point.
In summary, we thank the Reviewers for their efforts to improve our communication. We believe the above responses are fully responsive to their suggestions. In particular, we are sympathetic to Reviewer 2’s main point. However, we believe that fully addressing the response in the manner that the reviewer wishes is not scientifically pragmatic. We are open to consider further changes should the Editor feel it to be beneficial.